# Exploring Visualisation Methodology of Landscape Design on Rural Tourism in China

**Weijia Wang** [1,*], **Makoto Watanabe** [2], **Kenta Ono** [2] **and Donghong Zhou** [3]

[1] Graduate School of Engineering, Chiba University, Yayoi-cho 1-33, Inage-ku, Chiba 263-8522, Japan
[2] Division of Design Science, Faculty of Engineering, Chiba University, Yayoi-cho 1-33, Inage-ku, Chiba 263-8522, Japan; m.watanabe@faculty.chiba-u.jp (M.W.); k-ono@faculty.chiba-u.jp (K.O.)
[3] School of Design, China Academy of Art, Hangzhou 310002, China; 0106018@caa.edu.cn
[*] Correspondence: 19wd8301@student.gs.chiba-u.jp

**Abstract:** Rural tourism has become a hot topic in China in the context of the nation's rural revitalisation. Rural tourism allows tourists to experience local life and promotes local economic development. However, there is considerable controversy over the landscape design of ancient Chinese villages. Many problems, such as how to design and protect the landscape of these ancient villages and how to improve the tourist experience, are not resolved. For our research object, we selected the ancient Gaotiankeng Village in Kaihua County, Zhejiang Province. Using questionnaires, image interviews, and some user experience techniques such as mental maps, we collected user experience data by assessing design cases. The visualisation method presented a wide range of experience in the landscape and planning field. This study primarily used computer image processing, image entropy calculation, and colour mapping to process the data. A visualisation framework was defined to highlight the landscape aesthetics, landscape service, and tourists' emotion. The results indicated the relationship of three elements. The objective of our study was to develop a method of landscape design and planning that can effectively enhance tourists' experience and provide practical suggestions for rural landscapes and relatively better services.

**Keywords:** visualisation methodology; landscape design; tourists' experience; landscape aesthetics; entropy

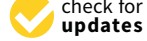



## 1. Introduction

The rural issues that China faces are a priority for the Chinese government. The development of local economy through tourism is one of the most promising approaches. Design and landscape planning methods provide an opportunity for the rural tourism industry to enhance tourist experience. According to People's Daily Online [1], from January to August 2020, the total number of local tourists travelling in rural places was 1.207 billion, which generated a total income of CNY 592.5 billion, an operating rate of 94.5%, and 10.61 million jobs in the industry. Although rural design and rural tourism has become a sought-after topic, the development and utilisation of ancient villages present some limitations. The landscape design and planning efficiency has not been completely realised, causing some wastage of resources and ecological destruction. Homogenisation or standardised design has affected many ancient villages in China, leading them to lose their original culture and traditional characteristics, leading them to become similar to a community within a city.

In the last six decades since 1960, considerable progress has been witnessed in landscape visualisation through the work of photography [2,3]. Several major technologies, such as eyes-tracking images, have been developed to determine user perception and experience. The literature has revealed that six mapping methods are predominantly available for visual landscape research [4]: compartment analysis, 3D landscapes, grid-cell analysis, visibility analysis, landscape metrics, and eye-tracking analysis. Entropy images

are used to connect visualisation with landscape aesthetics, which emphasise the subjective impressions of visual complexity in landscape scenes [5]. However, no specific visualisation framework is available for designers or planners to evaluate and improve tourist experience through the design method. Therefore, designers and planners cannot directly use the visualisation method to improve user experience through design projects.

According to Andrew Lovett [6], the visualisation method should answer three main questions: 'when'? (to use the visualisation method), 'what'? (to include the information), and 'how'? (to display the information). A standardised expression is provided for the visualisation method for such a framework. The following points summarise the main contributions of this study:

1. A visualisation method is proposed to evaluate user experience during an entire trip.

2. A new visualisation framework is proposed to analyse the correlation among emotion (user experience), landscape aesthetics (type), and landscape services (active and passive enjoyment), and the results show the models of this relationship.

3. The use the visualisation method is discussed.

The purpose of this study was to study and discuss the possibility of using visual methods to help design and planning for rural tourism user experience. A relatively complete visualisation framework was proposed using related user experience design methods. This framework was applied in the rural design and planning to render it highly practical. Simultaneously, the framework was used to improve tourist experience under the premise of protecting the culture of ancient villages.

## 2. Literature Review

### 2.1. Background of Ancient Villages

The rural landscape is to a large extent a historical product, integrating the positive factors of society and environment [7]. Providing a unique cultural value is indispensable and a core necessity for ancient villages. The Chinese government has promoted the rural land consolidation project from top to bottom at an unprecedented scale in recent years, vigorously developing tourism projects with the help of the excellent natural resources located in the countryside. The extension and expansion have continued in numerous villages, especially ancient villages [8].

An ancient village is similar to a museum: it contains treasures, and serves to convey regional culture and represent history. Therefore, an interdisciplinary exploration and practice is indispensable for current landscape designs. At the same time, dynamic thinking can reduce the hidden risks of the loss of ecological and cultural landscapes [9]. For example, it can help to determine approaches to design and offer service through tourist user experience. The relevant content of the design cooperates with the planning and design of the landscape to enhance the tourist experience. Such designs are creative [10], and can organically combine ecology and human perception and reflect a respect for the ecological environment [11].

The historical landscape of ancient villages in China is an essential element of such villages. Rural settlements, where people gather and settle down, are also space for people in the city and villages to gather. These landscapes are not merely rural buildings, but also public facilities for residents [12] and have their own cultural attributes. Consequently, a rural settlement is a community with architectural form and people: it contains various activities, economic activities, and active elements.

### 2.2. Landscape Service in Rural Places

To begin, two changes are critical in the context of the Anthropocene [13,14]. First, the emphasis on the ecological environment has become mainstream; second, the approach to utilise the full potential of the initiative of the people shaping the ecosystem through not only design, but also through economic and social development. The process and dynamics of designing and implementing nature-based solutions [15] include creating and using multidisciplinary and interdisciplinary knowledge, which helps reach solutions that

successfully balance economic, social, and ecological goals. Planning and conceptualisation must occur at various levels instead of on the basis of one-sided decisions [16].

Landscape service indicates that human activities, as well as biological and nonbiological factors, together shape the landscape. The interdisciplinary concept integrates additional services related to providing space [17]. Tiziano Cattaneo et al. [18] proposed the use of the design as a tool that would serve to revive the lost relationship between the environment, culture, heritage, and citizens to promote inclusive tourism as a means for social integration. Tourism promotes local development, while increased tourism infrastructure provides access to new areas. This new access positively affects the visitor rate [19]. Rural development is driven by rural tourism. With the recent increase in the number of tourists in ancient villages, many tourists have shown great expectations from ancient villages. From their visits, they can learn about the local people and traditional local culture. The wisdom of traditional culture cannot be ignored [20].

The landscape, being the core of tourist attractions, acts as the start and end factor which attracts tourists, prompting them to evaluate the experience as satisfactory [21]. Aesthetic perceptions and preferences can be location-specific for most people, based on local geographic and cultural characteristics, moral beliefs, life experience, and the use of specific areas. Therefore, this tour is a continuous superimposing of past personal experience with actual activities, and it depends on the individual.

Roy Haines-Young and Marion Potschin grouped landscape services [22] and landscape service (LS) classification under the Common International Classification of Ecosystem Services (CICES):

- Provisioning: nutrition, material, energy, daily activities;
- Regulation and maintenance: regulation of wastes, flow regulation, regulation of physical environment, regulation of biotic environment, regulation of the spatial structure;
- Cultural and social class: health, enjoyment, self-fulfilment (personal), social fulfilment.

'Passive enjoyment' is considered to be an attractive atmosphere that does not require human intervention. Created by nature, this atmosphere may be a quiet place for reading or the opportunity to see what cannot be seen in the city: wild animals or cultural, historic landscapes, and heritage structures [23], providing human aesthetic appreciation. On the other hand, 'active enjoyment' gives visitors either the possibility of interacting with the landscape or participating in the atmosphere. In this case, the distance between humans and the landscape is reduced. This allows visitors to enjoy time with nature, such as through hiking, swimming, gardening, hunting, or fishing and provides children play and recreational opportunities, such as tourism, ecotourism, and research activities [24]. If tourism is considered as a means of social inclusion and resource, social innovation in the field of community development can be interpreted as modifying the process of planning, governance, and design in order to improve the participation of communities, residents, and tourists. The focus is then on a collaborative and innovative society in the context of rural communities.

*2.3. Visualization Methods Based on Landscape Atheistic*

Various forms of visual communication have a long history in environmental management (Table 1), especially in landscape architecture and planning [25].

Visualisation has a dual advantage: it can realistically improve virtual viewing ability and the degree of interaction while significantly reducing related costs. The use of visualisation as a participatory tool continues to expand. The opportunities and demands for visualisation are increasing with the participation of members of the public and the development of the landscape planning field, covering a more comprehensive range of issues. Beatrice John et al. used decision-visualisation environments to obtain an active development network and form an efficient discussion community [26]. They also organised activities and shared their indirections, made suggestions for design and planning, and determined whether the differences in viewing behaviour were complicated [2]. The differ-

ence in degree is relative, while the complexity is represented by the spectral entropy of the photo. The eye-tracking experiment serves to measure visual behaviour when observing photos. More urbanised landscapes offer more extensive and decentralised exploration [27].

**Table 1.** Development of visualisation in the landscape.

| Years | Methodology | Literature |
|---|---|---|
| 1960 | Photographs and photomontages have been widely used. | [2,3] |
| 1990 | CAD, GIS and landscape visualization software enhanced the possibilities for digital representation. | [28] |
| In the past decade | The availability of free virtual globe software has opened up additional opportunities for real-time display, particularly given the scope for customization and incorporation of 3D buildings or vegetation. | [29] |
| Currently | CAD or GIS database and then generate three main types of 3D outputs. 1. Rendered still images (or scrolling panoramas) from defined viewpoints, 2. animatedsequences (showingfly-throughs along specified paths or changes over time), 3. real-time models (or virtual worlds) where the user has the ability to freely navigate a landscape. | [6,28] |

Using the entropy methods also combines different approach used in order to analyse a specific scene. Pearson's correlation coefficients calculated the eye-tracking metrics and the inverse function of the spectral entropy. These results confirm the findings of the analysis performed on the different urbanization classes, as they also indicate an increase in visual exploration when the visual complexity of the landscape image increases. Entropy is relevant to applications in environmental aesthetics [30]. Lien Dupont et al. computed if differences in viewing behaviour are related to differences in complexity, expressed by the photograph's spectral entropy [27]. Numerous studies have shown that image entropy is strongly correlated with rated visual diversity and complexity. Consequently, entropy strongly indicates subjective impressions of visual complexity in landscape scenes. Stamps mentioned that numerous studies have demonstrated image entropy to be highly correlated with rated visual diversity and with visual complexity [31]. Entropy, thus, strongly indicates subjective impressions of visual complexity in landscape scenes [5]. This conclusion is relevant for our study, which is concerned with the visual complexity of a landscape view as perceived and experienced by people.

## 3. Case Study Selection and Survey Method

### 3.1. Gaotiankeng Village

The Gaotiankeng Village (Figure 1) is located in Quzhou, in western Zhejiang Province, at an altitude of approximately 700 m and has a history spanning approximately 800 years. Here, the average temperature in summer is approximately 5 °C lower than that in the surrounding areas, making it a natural summer tourist spot. Although the mountain peak is not high, Shan Lan is surrounded by beautiful scenery and blue sky. It is also the best point to observe starry sky in the Donghua area. The Gaotiankeng Village is a typical ancient village from the area south of the Yangtze River. The buildings in the village mainly consist of rammed earth structures, with various shapes and spectacular views. The villagers still retain a simple lifestyle of digging ponds in front of their own houses to feed freshwater fish, a feature that has excellent development potential.

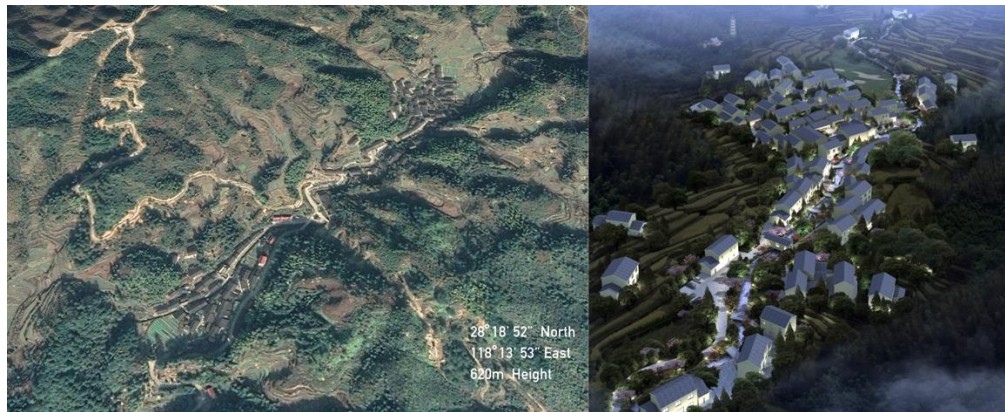

**Figure 1.** Location and the panoramic rendering of Gaotiankeng Village. Google map link: https://www.google.com/maps/@29.3125082,118.2288567,966m/data=!3m1!1e3!5m1!1e4 (accessed on 12 December 2017).

### 3.2. Design Goals and Outcomes

Four design goals have been defined based on the planning and designing project: building and design, environment, community inclusion, and economic growth. We established different design outputs across the project. As shown in Figure 2, for the design and planning of the Gaotiankeng Village, we created 30 nodes in the main tour route. The output is the concrete expression of the four set goals, and the overall design and planning of the landscape's appearance, location, material, service content, etc. were performed. Figure 2 clearly indicates that the design output of the four goals overlaps to a certain extent. For example, the improvement of community tolerance is closely linked to landscape design and environment creation. The rural does not need high-cost design; instead, it requires flexible design and a layout with extensible functions.

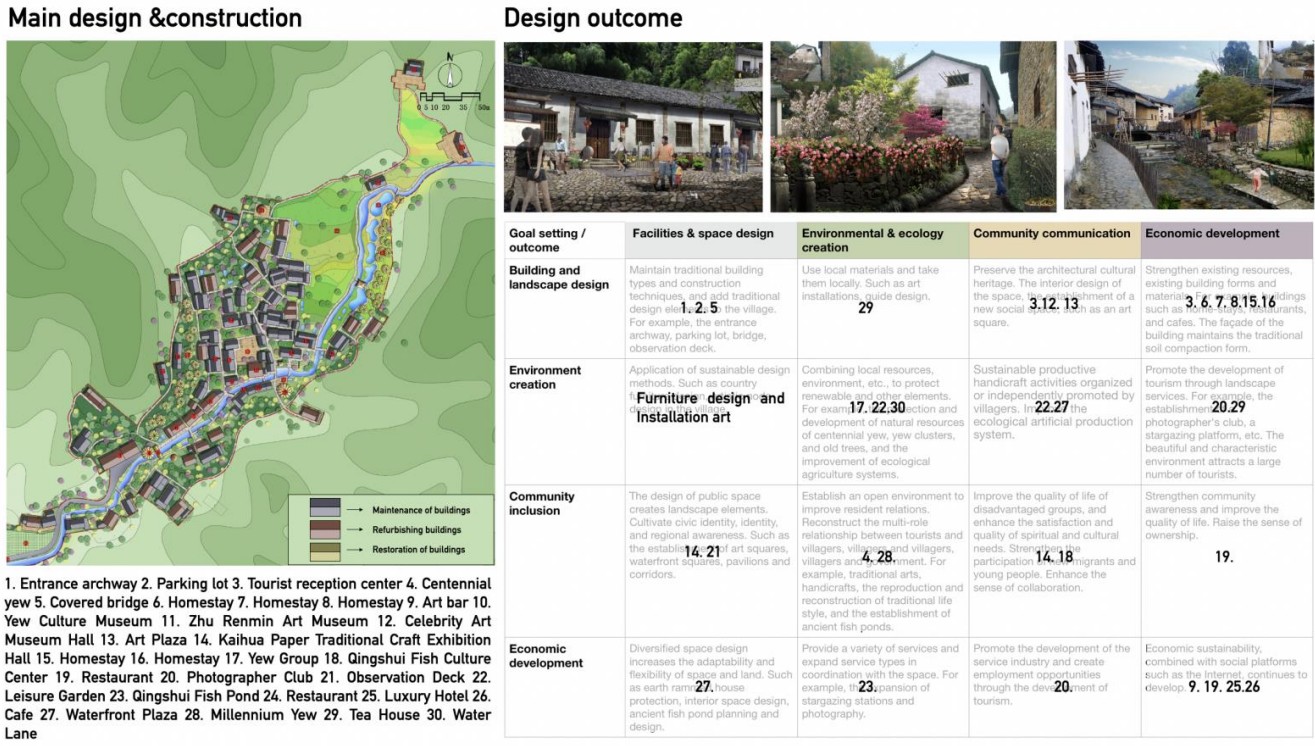

**Figure 2.** Design and planning of project to renew the Gaotiankeng Village.

### 3.3. Survey Method

To understand the user's landscape and service preferences, we used questionnaires, mental maps, and photo interviews. Furthermore, we took a tour and conducted a recording to produce a comprehensive view of the user's perception and aesthetics. The tourist's perception of the landscape and their preferences when traveling are highly complicated and can be disturbed by many external substances. Therefore, our approach involved a timely inquiry. Photos provide a visual stimulation that matches real life. The landscape experience is positive, since the photos holistically show the landscape [32]. Due to this, perception-based assessment is highly reliable and is suitable for capturing people's landscape preferences [33]. Measuring the biophysical characteristics of the landscape and its spatial layout allows us to quantitatively describe the visual characteristics of the landscape [34].

Along with this, we also used an image questionnaire. A total of 20 tourists participated in our survey of questionnaires and interviews. The female participants outnumbered the male participants (56.8% vs. 43.2%). The questionnaires were based on a Likert scale, 5-point with indicating 'not at all' and indicating 5 'very much'. Among the 32 tourists who followed the survey, 26 took the same route (as the main tour route) (Figure 3), and the 20 tourists who participated were from this group. The survey was an image questionnaire, with a total of 12 equidistant nodes for tourists to score and evaluate. Since many buildings served as the display or experience spaces of traditional culture, the types of services provided by the landscapes were different. We classified these according to the enjoyment of landscape service, but the service content level—such as service attitude and service quality—were not considered in this study. The interviews served to collect as much comprehensive information about user experience as possible.

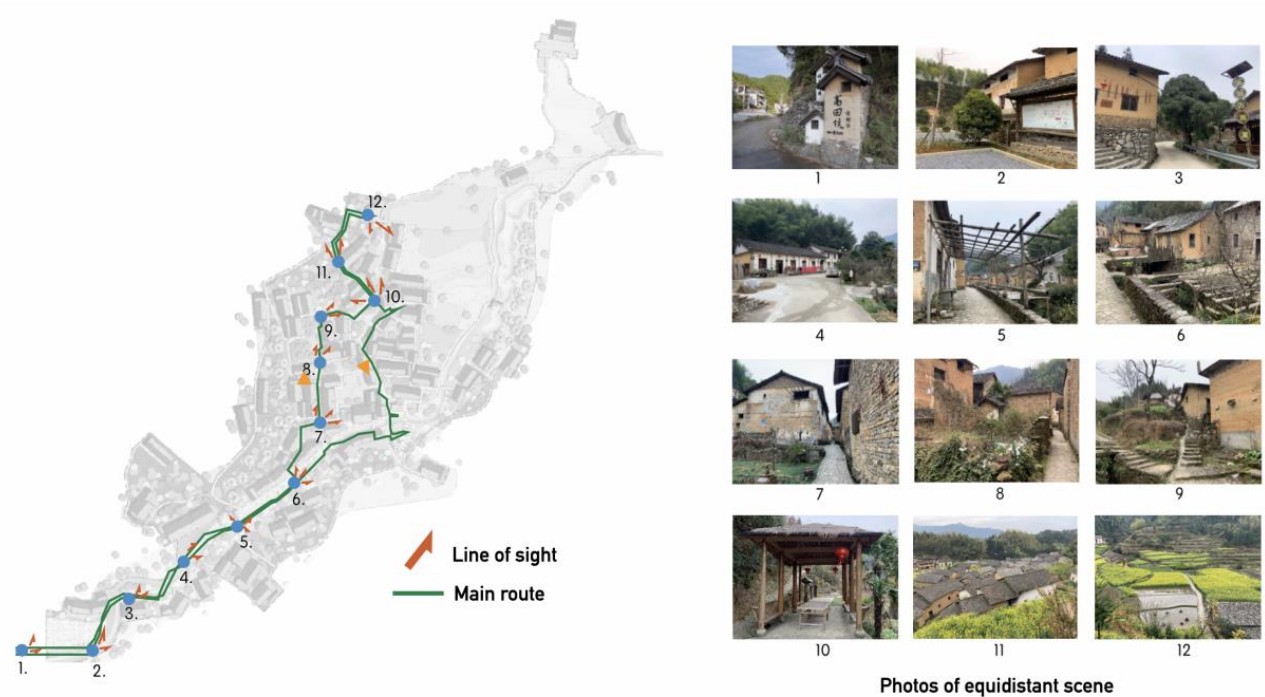

**Figure 3.** Main tour route, line of sight, and photos of equidistant scene.

## 4. Method

### 4.1. Design Process

1. We analysed 12 equidistant scenes along Path 1 (100 m) (photos by teams), shooting the subject and the direction of the line of sight in the spring of 2021. Since humans mainly experience the environment visually, the understanding of environmental experience requires visual materials. Inviting people to evaluate an actual landscape on the spot is

expensive and time-consuming. It also limits the types of landscapes that can be studied by restricting the location; thus, using a specific location and case is a suitable choice. Concerns about the representation validity of photos in visual landscape assessment can be found in the literature. For example, Marjanne Sevenant compares three stimuli: in-situ landscape, panoramic, and standard normal photos. The site landscape in this study was evaluated during the field visit, indicating that standard formal photos were more suitable for measuring landscape preference variables [35]. Each point in a total of 12 equidistant points was measured on the mainline and photographed. The measurement started from the village entrance and ended with the viewing platform on the top of the mountain.

2. Computer image processing and the depth of field was utilised to distinguish between the foreground, middle ground, and background [36]. The foreground and background elements in the visualisation play important roles in determining the tourists' visual preferences, but the middle ground is key [36].

3. The picture was processed using image entropy processing. The formation of image entropy is as follows:

$$H = -\sum_i \sum_j P_{i,j} \lg P_{i,j} \tag{1}$$

The processing has two steps (Figure 4): the first step is the grayscale calculation for the entropy of the photo, while the second step is to superimpose the original image and use the rainbow colour template to map the entropy image to obtain the final 10-pixel image. An entropy image is a collection of the entropy values of every image patch [37]. An entropy image is a collection of the entropy value of every image patch. (Signal Processing: Image Communication, 2018).

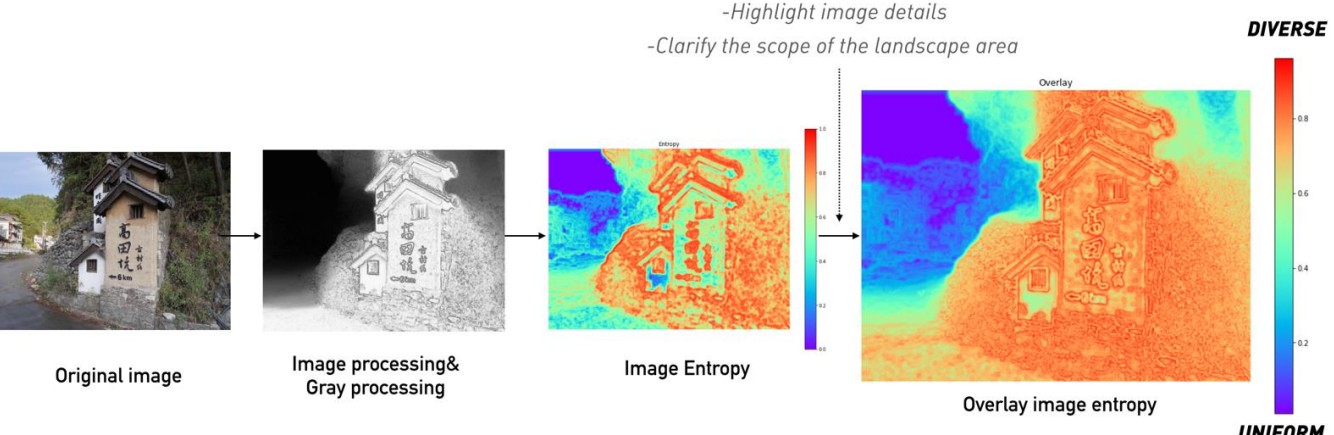

**Figure 4.** Flow table of the entropy image framework.

Based on the entropy image method, a tool was developed to calculate the entropy of image and to map the colour on the image (https://colab.research.google.com/drive/1jHJvCCV885arTxdvz4rMZwfUOml9oeOz, accessed on 28 September 2021).

*4.2. Visualization Framework*

1. Landscape aesthetics perspective: coherence, complexity [38]. Visual characteristics of the landscape: enclosure, legibility. Landscape space scale: hierarchy [39]. These listed factors of landscape aesthetics were analysed from the entropy images (Figure 5).

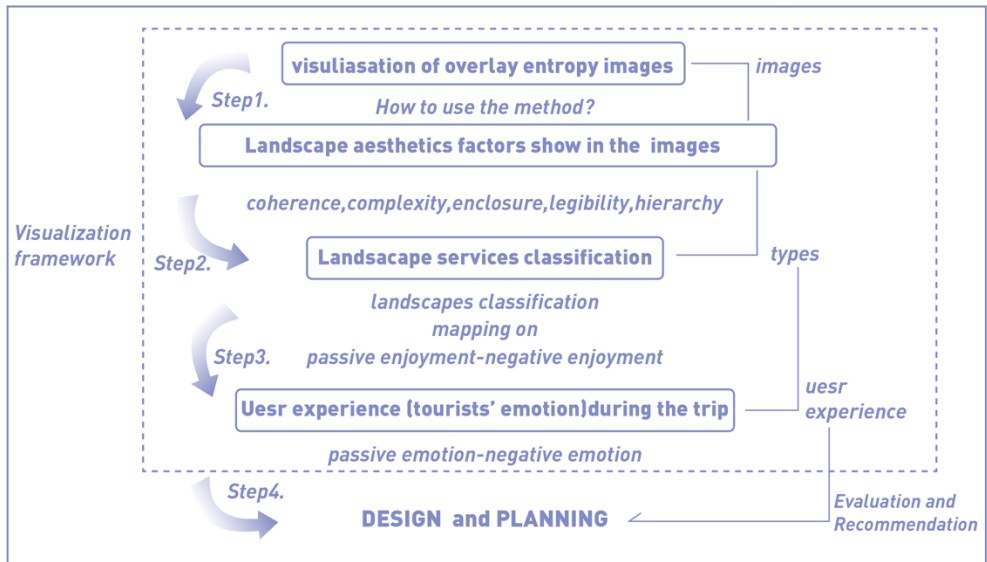

**Figure 5.** Flow table of the visualization method framework.

2. In Figure 5, we selected the fourth enjoyment [24] in the landscape service (LS), and divided it into two categories of passive enjoyment (PE) and active enjoyment (AE). PE includes aesthetic appreciation, values, and heritage, while AE includes recreation, tourism, and ecotourism. The HLE categories are modified and adjusted according to the actual situation of the Gaotiankeng Village (Table 2): agriculture, forestry, mountain, traffic, mining, processing of food and materials, building type, religion, fishery, and hunting.

**Table 2.** Enjoyment of LS&HLE in Gaotiankeng village.

| LS | HLE | Agriculture | Forestry and Mountain | Traffic | Mining | Processing of Food and Materials | Building Type | Religion | Fishery and Hunting |
|---|---|---|---|---|---|---|---|---|---|
| Enjoyment | PE | O | | O | | O | O | | O |
| | AE | | O | | O | O | O | O | O |

"O" indicates that the landscape component belongs to the PE or PA.

3. The sentiment of rural tourism is an important indicator to evaluate user experience. Tourists with a higher degree of pleasure in general show higher satisfaction and better behavioural intentions [40,41]. The role of emotion in leisure tourism research has been recognised in recent years. The travel experience usually includes feelings of satisfaction and pleasure [42–44]. Previous research has shown that emotions experienced affect tourist satisfaction [45,46]. To evaluate the tourism of a destination, satisfaction is essentially evaluated from four metrics: overall satisfaction, experience being in-line with expectations, wise decision-making, and experience being worthwhile. These reflect the comprehensive evaluation of the user's play experience (Figure 5). Emotions can also influence the decision to purchase travel and leisure services [47]. The assessments based on aesthetic preferences provided by tourists can be used as a reasonable basis for drawing entertainment demand maps. This method has multiple advantages: it is fast, efficient, and easy to replicate in other areas. The framework developed can be used as an input to support landscape management, identify areas most required by society, and quantify the demand for space recreation to support political strategies [48]. In addition, novelty is a pronounced emotional dimension for tourists visiting ancient villages. Novelty refers to a psychological feeling of newness that results from a new experience. It is one type of potential construct of the memorable tourism experience [49,50]. In this paper, the following four measures are

selected as the evaluation dimensions and questionnaire design of the Gaotiankeng Village (Figure 6):

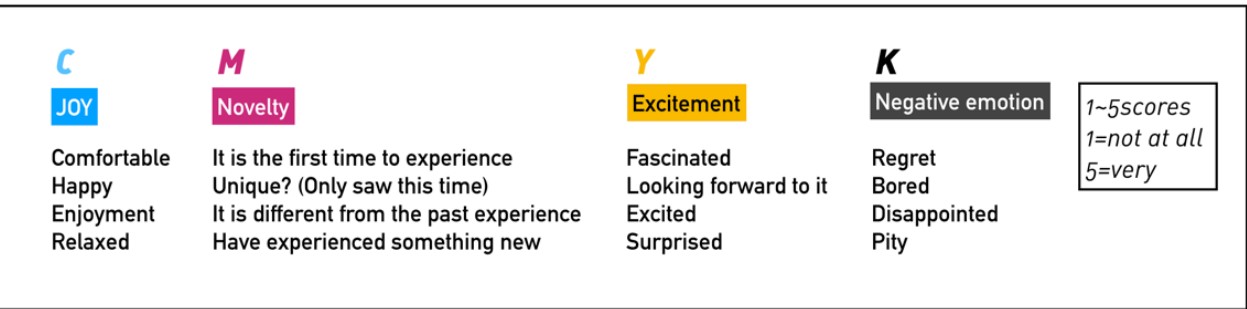

**Figure 6.** Evaluation dimensions of emotion in questionnaire.

The image questionnaire survey method measured the negative and positive emotions of the tourists according to the travel psychology evaluation scale. The four dimensions are defined by the colour dimension CMYK (Figure 6). The survey results confirmed the primary hue according to two parts with the highest scores.

4. As shown in Figure 7, we placed the 'emotion', 'landscape atheistic factors', and 'landscape service' parameters in a cube. The colour hue represents the emotion degree. The entropy image shows that the landscape atheistic forms coherence, complexity, enclosure, legibility, and hierarchy. For the service element according to the entropy image result, the range is '0–1'. We split the whole value into three equal parts, and the parts with the highest and lowest values were excluded. The adjacent values of three equal points, 0.83, 0.49, 0.17, which can be clear distinction in the visual ground, were taken. The colour bar contains the information of landscape service classification and landscape components (Appendix A).

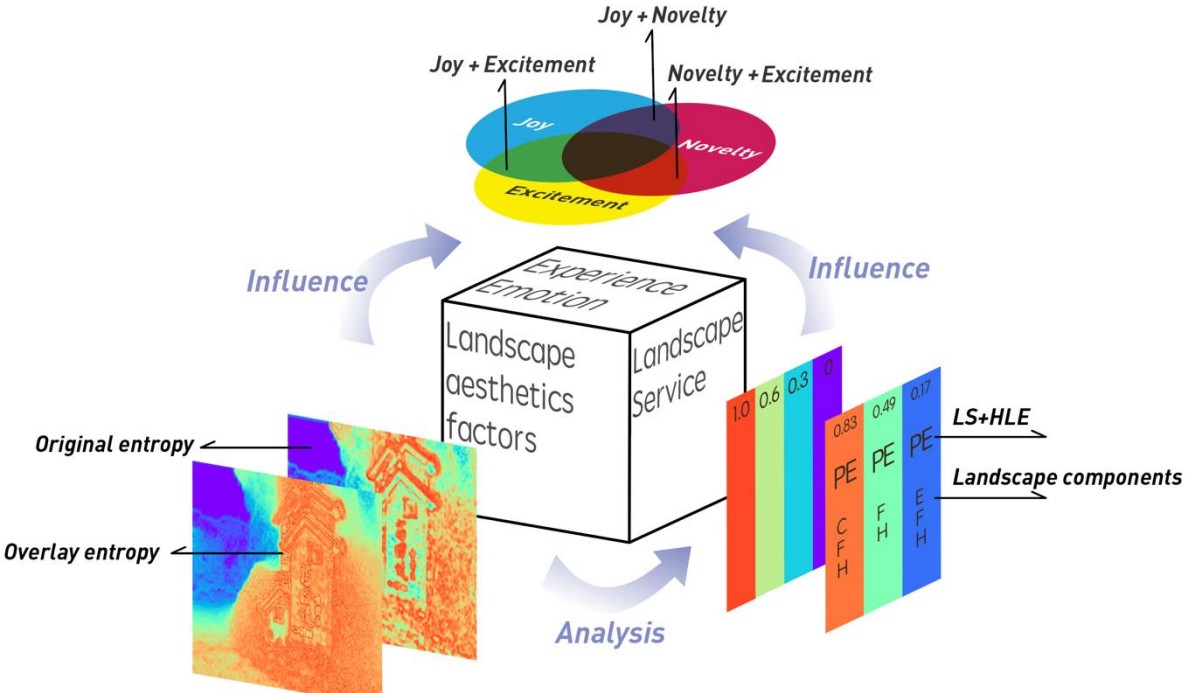

**Figure 7.** Visualisation framework cube.

5. A multiple linear regression model was used to explain the linear relationship model of variables. The tool used was spss.

6. This framework cube could give the designers and planners practical recommendations during the design project.

## 5. Result

### 5.1. Identity of Visualisation

5.1.1. Landscape Aesthetics Perspective: Coherence, Complexity

Åsa Ode et al. [38] used three dimensions to describe the visual landscape characteristics: (1) the distribution of landscape elements, describing the richness and diversity of elements in the landscape; (2) the spatial organisation of landscape elements, describing the arrangement of different components in space; and (3) the changes and shapes of elements and patterns, describing the degree of changes in the landscape and the shapes of elements and patterns. Research has shown that complexity and coherence are closely linked concepts for the landscape experience of tourists [51]. Metrics for the arrangement of units and the degree of repetition in the landscape are essential for describing the consistency associated with complexity [38]. Therefore, capturing the changes and shapes of the landscape space, organisational elements and patterns are essential. Figure 8 shows that the indicators of the arrangement of units in the landscape and the degree of repetition are high. This is especially true for the architectural landscape and the road landscape, which appear up to 9–10 times out of 12 times, giving continuity to the visitor experience.

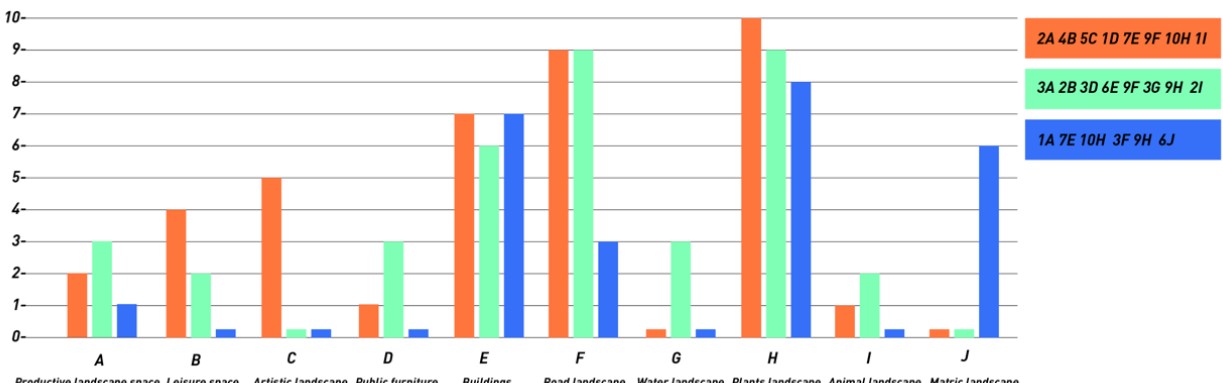

**Figure 8.** The types of landscape are spread over three different areas.

The degree of openness of the edge of each unit space and the relationship between visual and physical continuity was assessed. The visually walkable open area is relatively fuzzy in the village. First-time tourists can explore the place on their own, for example by following field trails. The staggered location of the houses increases the rhythm of the experience and adds a sense of fluency to the continuous landscape. The open area of sight directly reflects the immediate senses.

The complexity indicators are as follows: 1. the distribution of the landscape attributes, richness, and diversity; 2. the spatial organisation of the landscape attributes, edge density, heterogeneity, and aggregation; and 3. change and contrast are important, such as contrast, shape change, and size change.

Complexity appears in the development of indicators for several landscape functions, including visual quality and biodiversity. Complexity has been used in environmental psychology as an explanatory factor for landscape preference. More information can be found in more complex landscape photographs [20].

Variables such as coherence, complexity, disturbance, visual scale, or naturalness overlap with aesthetic qualities. These can be used to identify the aesthetic preferences of users for outdoor environments. Additionally, aesthetic values are typically closely related to recreational ecosystem services since the aesthetic enjoyment of the landscape is a factor contributing to the choice of location for conducting recreational activities [52].

### 5.1.2. Visual Characteristics of the Landscape: Enclosure, Legibility

As shown in Figure 9, we classified the 12 pictures into types of enclosures. According to the classification and description of Robinson's permeability of enclosure [53], we classified the enclosures in the pictures as visually enclosed, physically enclosed/visually enclosed, physically open/visually open, physically enclosed/visually open, and physically open. We connected the type to each figure and marked it according to the composition form of each figure. The first 10 pictures represent the nodes before the summit and depict the process of traversing the village, with type one–three interspersed with each other; 11–12 are type four, located at the top of the mountain and looking down at the landscape, with a bird's eye view of the village.

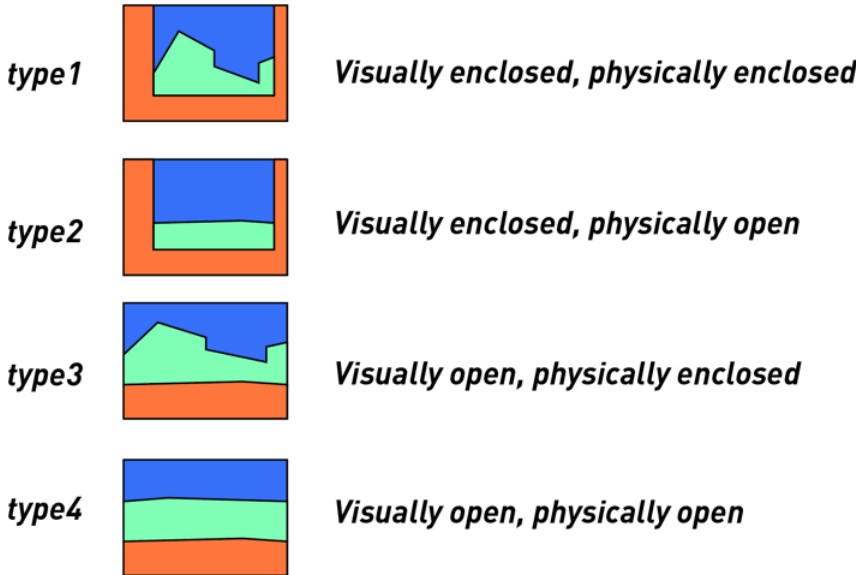

**Figure 9.** Enclosure type of the visual based on the entropy images.

Traditional Chinese courtyards are typically fully enclosed by walls and/or architecture to make the associated properties private. Nowadays, buildings are less about housing and more about creating public spaces that provide services. The landscape is becoming increasingly important, rather than the architectural components, and is now often used to shape open spaces. The clarity of the village landscape contours, the coherence, and the integrity of the village spatial structure and form are the key points to improve the visual quality and enhance the attractiveness of the village. The computer image processing used in this study can reflect the user's visual impression of the landscape more realistically, intuitively, and clearly. The details of the final visual image can be clearly expressed through the overlay of the depth of field and details, and the clear expression of texture and contour is obvious.

### 5.1.3. Landscape Space Scale: Hierarchy

The landscape elements themselves have an impact on the provision of benefits, their context, the relationships between them, and their spatial arrangement placed in the spatial scale section [39]. Tongji University suggests that 'spatial confinement' is the characteristic of the degree to which the viewer encloses elements of landscape space, and that it is mainly influenced by the vertical factors of space within the viewer's spatial context. Spatial confinement is characterised by both openness and closure. The spatial interface scale (spatial length) is as follows: <8 m; 8–25 m; 25–110 m; >110 m. The visual interface scale (sight distance) is as follows: <8 m; 8–25 m; 25–110 m; 110–390 m; >390 m. People typically have a psychological tendency to want to enter another type of space immediately after passing through one type of space [54]. Foreground and background elements in visualization play an important role in determining visual preferences, but the middle

ground is the most important [34]. As shown in the Figure 7, there are four levels from 0 to 1. We removed the highest and lowest values, where the lowest value represented the mid-play and sky views and the highest value represented the closest distance and complex changing views, such as rocks and leaves. As shown in the image, the middle region is divided into three parts: orange, green, and blue. According to the HLE classification, we mapped the landscape types in three different regions.

### 5.2. Rhythm of Changes in Emotions

First, we defined the colours for the four emotions: joy, novelty, excitement, and negative emotion, according to which we could quickly identify the colours after superimposition. As shown in the diagram, joy + novelty = purple tones, joy + excitement = green tones, novelty + excitement = red tones. As three colours are superimposed, the primary colour is formed by the superimposition of the two higher value colours, while the third colour acts as a blend in a particular proportion. Large and well-prepared paths are highly used for recreational activities, but they are visually less appreciated than nature trails [55]. Most visitors chose to walk down the countryside trails and freely pass between buildings. Therefore, we selected and designed the main route with 12 nodes based on several studies.

As the figure shows (Figure 10), when the colour of each node varies, i.e., the mood of each node is different. The eighth is the turning point of the whole tour mood, and the first to seventh are dominated by joy and novelty, as demonstrated by blue and purple shades. The next four are dominated by joy and excitement, with a greenish shade. Constantly different visual and service experiences through the design and planning of the landscape are necessary if the tourists are to maintain their sense of pleasure throughout.

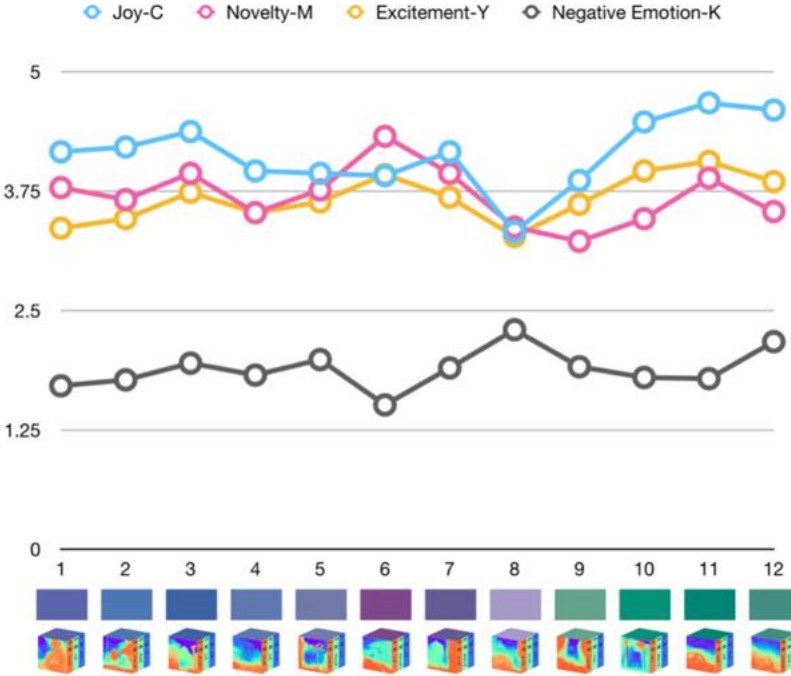

**Figure 10.** Result of emotion questionnaire and the colour definition of visualisation.

Designers and planners should consider the novelty of emotions of the tourists during the middle area when planning the entire process. For example, the novelty score increases at the sixth node thanks to the local cultural landscape: ancient fish farming. Fish farming in running water pit ponds has been performed for 600 years in the Gaotiankeng Village. Every family has a fishpond, and they raise fish mainly to meet their own daily consumption. If there are any extra fish, these are sold in the market. The ancient method of fish farming in ponds has two main features: water is drawn from the mountain spring and fish are fed

on the grass in the mountains. The fishponds (Figure 11) in the village are built along the mountain, with water coming in from high places and water coming out from low places so that the grass carp are not muddy and, as a result, taste better. This element of the human landscape is unique to the village; therefore, it is new and exciting for foreign visitors. The uniqueness of the landscape highlights its difference from other regions, and the creation of differentiation is also an essential topic of discussion in village planning.

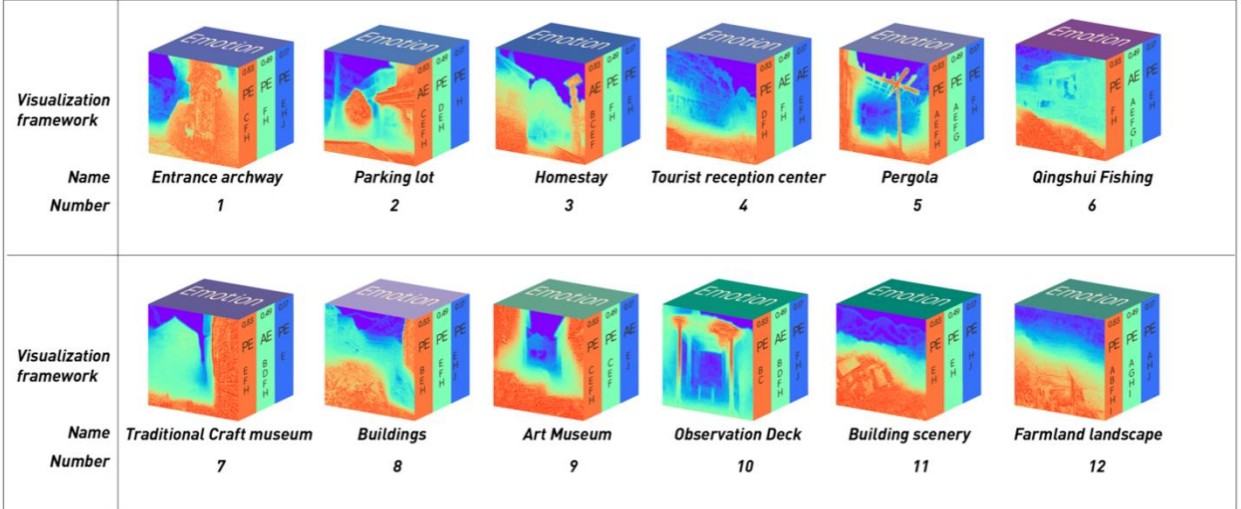

**Figure 11.** The visualisation of the framework during the whole trip.

A small amount of boredom and aesthetic fatigue is expected at the eighth node, as well as a reduced mood, due to the constant shuttling through the ancient village. Moreover, since the interview time was in early spring, everything was still in the recovery stage, and a visually aesthetic landscape had not yet been formed. To maintain a high sense of experience however, we placed art installations on the roadside during the design. Since most of them were made with flowers and food items (such as corn, chilli, and other dried food) used as raw materials for the design, many were damaged due to the seasonal effects. Despite this, the visitors were surprised and delighted to find that they were still intact during the tour and stopped to take pictures of them.

Visitors already have complacency when they reached the last four nodes, and they were ascending the mountain. When they were about to reach the top to enjoy the scenery, their mood was primarily pleasant and excited. The pleasure value reached its peak when they arrived at the viewing platform, i.e., the eleventh node. When colours are used to display the value of the emotion experienced, the visual effect created gives designers and planners a much stronger direct perception than simple data. In contrast, the change of colour rhythm can show the user's emotional experience during the whole tour so that the design and planning of the node can quickly give relevant suggestions.

### 5.3. Relevance Analysis

Table 3 indicates that the correlation coefficient between the enclosure type and stratification type is 0.2, which in turn indicates that the two values have a positive correlation. The correlation coefficient value shows that the correlation between the two is not particularly strong. The correlation coefficient between circumscribed type and emotion is 0.316, and the correlation between emotion and circumscribed type is stronger as compared to stratified type, also showing a positive correlation. The correlation between stratification type and emotion is the highest among the three correlations, with a value of 0.392, indicating that the two show a positive correlation.

**Table 3.** Relevance of enclosure type, service in ground type and emotion.

|  | Enclosure Type | Service in Ground Type | Emotion |
|---|---|---|---|
| Enclosure type | 1 | 0.200 | 0.316 |
| Service in ground type | 0.200 | 1 | 0.392 |
| Emotion | 0.316 | 0.392 | 1 |

### 5.4. The Influence of the Service Type in Different Areas on the Emotion

The least-squares estimation method is used in this paper to estimate the model parameters. This method is the most commonly used classical estimation method, and its principle is to reduce the deviation between the observed values and the regression values as much as possible. Additionally, it provides an integrated consideration of all samples of the value of the deviation, and the requested will receive the parameters of the least-squares estimation. The requested parameters can be obtained from the empirical regression equation (formula):

$$\hat{Y} = \hat{\beta}_0 + \sum_{i=1}^{n} \hat{\beta}_i X_i \tag{2}$$

We could not definitely conclude that the explanatory variable $Y$ is linearly related to the independent variable $X_i$ in the actual problem. Fitting the relationship between the explanatory variable $Y$ and the independent variable $X_i$ with a linear regression model before parameter estimation is an assumption made based on qualitative analysis. The following three tests are used in this paper: F-test for the significance of the regression equation, $t$-test for the significance of the regression coefficients, and the test for the goodness of fit.

#### 5.4.1. Effects on Positive Emotions

The model summary in Table 4 shows that the decidable coefficient R2 of the equation is 0.27 and the value of the amount of change in F is 0.984. Meanwhile, the Durbin–Watson (DW) value is considerably small at 1.683, which is relatively close to 2 and can be roughly considered as not having an autocorrelation. These values indicate that the model fits reasonably well. The impact regression coefficients are estimated below. Table 4 presents the results of the regression coefficient test for the equation. It shows that the $p$-values of the significance levels of the independent variables and constants in the equation are greater than 0.05. This, in turn, indicates that the regression coefficients of the model failed the significance test and that the linear relationship with the dependent variable negative emotion is not significant. This implies that the effect of stratification type on positive emotions (Y2) is not significant, the tolerance of all variables is less than 5, and the variance inflation factor VIF values are close to 1, indicating that there is no multicollinearity.

**Table 4.** Model summary.

| Model | R | R Equation | Errors in Standard Estimation | F | Durbin–Watson |
|---|---|---|---|---|---|
| 1 | 0.519a | 0.270 | 0.24206 | 0.984 | 1.683 |

The effect of stratified type variables on positive emotions is not significant, as shown in the results of the regression coefficient test (Table 5). The regression coefficients of the two variables are negative, except for the blue area, where all regression coefficients are positive. This shows that the blue area has a positive effect on positive emotions, while both the orange area and the green area have a reverse effect relationship. The largest regression coefficient is 0.279 for the blue area, −0.154 for the green area, and −0.077 for the orange area. The blue area has consequently the greatest effect on positive emotions, while the orange area has the least effect on positive emotions. A simple multiple linear

regression model can be constructed based on the regression coefficients, and the results of the obtained regression equation are shown below:

**Table 5.** Parameter estimation.

| Model | | Unstandardised Coefficient B | Standard Error | Standardisation Coefficient Beta | t | Significance | Covariance Statistics Tolerances | VIF |
|---|---|---|---|---|---|---|---|---|
| | (Constants) | 3.773 | 0.291 | | 12.956 | 0.000 | | |
| 1 | Type in orange area | −0.077 | 0.200 | −0.124 | −0.385 | 0.710 | 0.877 | 1.140 |
| | Type in green area | −0.154 | 0.157 | −0.314 | −0.982 | 0.355 | 0.891 | 1.123 |
| | Type in blue area | 0.279 | 0.192 | 0.450 | 1.451 | 0.185 | 0.950 | 1.053 |

Dependent variable: positive emotion.

Positive emotion = 3.773 − 0.077 × type of orange area − 0.154 × type of green area + 0.279 × type of blue area.

5.4.2. Effects on Passive Emotions

The model summary in Table 6 shows the goodness-of-fit test of the equation and provides the summary information related to the equation model. The table indicates that all variables are enter into the regression equation as required in the regression model equation, and the decidable coefficient $R2$ of the equation is 0.248 while the value of the amount of change in F is 0.491. Moreover, the Durbin–Watson (DW) value is considerably small at 1.911, which is highly close to 2, indicating that there is no autocorrelation. This indicates that the model fits reasonably well. The impact regression coefficients are estimated below.

**Table 6.** Model summary.

| Model | R | R Equation | Errors in Standard Estimation | F | Durbin–Watson |
|---|---|---|---|---|---|
| 1 | 0.498a | 0.248 | 0.2108428 | 0.491 | 1.911 |

Table 6 represents the multinomial results of the regression coefficient test of the equation. It shows that the significance levels of the independent variables in the equation have *p*-values greater than 0.05, indicating that the regression coefficients of the model failed the significance test. This also explains that the linear relationship with the dependent variable negative emotion is not significant, i.e., there is no significant effect of stratification type on negative emotion (Y1). The tolerance of all variables is less than 5 and the variance inflation factor VIF values are close to 1, indicating that there is no multicollinearity.

The results of the regression coefficient test indicate that the effect of stratified type variables on negative emotions is not significant, and all regression coefficients are positive. This suggests a positive effect on negative emotions. The regression coefficient is 0.117 for the orange area, 0.220 for the green area, and 0.001 for the blue area (Table 7). The green area has a greater effect on negative emotions than the orange area, and the orange area has a greater effect than the blue area. A simple multiple linear regression model can be constructed based on the regression coefficients, and the results obtained are shown below:

Passive emotion = 1.642 + 0.117 × type of orange area + 0.22 × type of green area + 0.001 × type of blue area

**Table 7.** Parameter estimation.

| Model | | Unstandardised Coefficient B | Standard Error | Standardisation Coefficient Beta | t | Significance | Covariance Statistics B |
|---|---|---|---|---|---|---|---|
| | (Constants) | 1.642 | 0.254 | | 6.474 | 0.000 | |
| 1 | Type in orange area | 0.117 | 0.174 | 0.219 | 0.670 | 0.522 | 0.877 | 1.140 |
| | Type in green area | 0.220 | 0.137 | 0.522 | 1.606 | 0.147 | 0.891 | 1.123 |
| | Type in blue area | 0.001 | 0.168 | 0.001 | 0.004 | 0.997 | 0.950 | 1.053 |

Dependent variable: negative emotions.

## 6. Discussion

When producing mental maps with the tourists, our team found that they were highly cautious in selecting their destinations. They have a basic understanding of the local landscape features, culture, etc. Whether through articles, social media, or advertisements, they get a general image of the characteristics of the village before visiting it. They also maintain a sense of anticipation and curiosity. The results show that the type of service in the blue area has a positive effect on positive emotions, while the type of service in both the green and orange areas has a reverse effect on them. Since the trip scenario in the Gaotiankeng Village can be similar to the hiking scene, the tourists had expectations along these lines. Therefore, the planning and design of the type of blue area is often easy to ignore, whereas the environment and the landscape or services affect the emotions during the trip.

Emotional visualisation is a more intuitive and straightforward way of expression. Figure 12 shows that it has three main advantages: to improve efficiency, to improve satisfaction, and to reduce risks. Risk reduction here refers to reducing subjective errors and defining rural design and planning. These elements help bring better life experience scenes to villagers and tourists. The main functions of emotional visualisation for designers and planners are to record, reference, and suggest. These functions lie in the four stages of the involvement of art design in the countryside: cognition, expression, perception, and development. Different application emphases for design and planning are applied in the various stages. In the first stage and fourth stage, design and planning need to be balanced; in the second stage, the focus needs to be on design, with an emphasis on the design of landscape, products, and space; in the third stage, planning needs to be emphasised, with a focus on the creation of atmosphere, the improvement of users' visiting experience and the setting up of nodes. Planning and design have different priorities at different stages, and the use of emotion visualisation in different locations is also further necessary; hence, we should pay attention to the specific practice and application at the various stages.

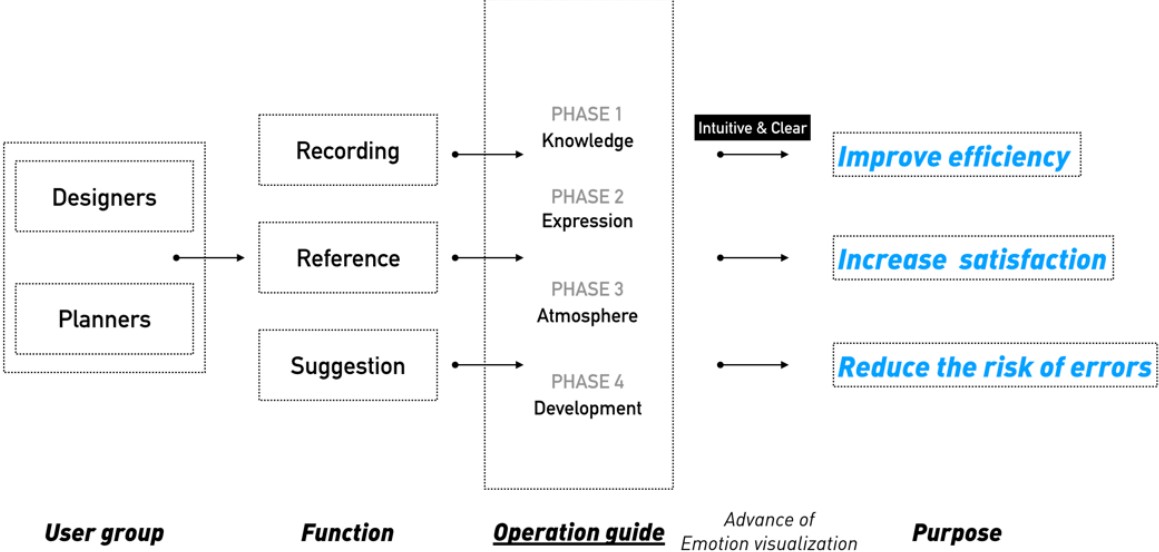

**Figure 12.** Visualisation methodology operation guide and purpose.

The landscape visualisation approach should be customised to the stages of planning. The intended purpose, audience, and resources should influence the content and presentation. Stakeholder involvement in visualisation design can improve communication efficiency. Robust empirical research is needed to better compare visualisation options. Visualisation methods require a structured evaluation environment in a realistic visual presentation.

In more conceptual terms, visualisations represent a mechanism to support the 'boundary management' functions [56] as crucial for the creation of knowledge systems underpinning sustainable development. However, assessing aesthetic values remains challenging, since they are only partially defined by the biophysical characteristics of the landscape that can be quantitatively described (e.g., vegetation type, spatial pattern) [57,58]. Hence, the limitations of this visualization method are difficult to assess all the aesthetic values, but this method can assess the main factors of aesthetics. The visualization framework needs to process one step by step, there were different results would appear if one of the steps' results are different. Mostly, the mental factors and perception factors depend on the individual. Therefore, the topic needs further thought.

First, the quantification of landscape services assessment needs to be given importance. The assessment of landscape services essentially deals with the complex and dynamic relationship between humans and the environment. Quantification helps planners and designers solve real problems and promote the ecological development of rural tourism.

Second, the relationship between emotion and landscape needs to be explored. Beautiful environments are dependent on human perception and aesthetics, and the environment is the ambiance that people can perceive. Therefore, quantifying the atmosphere and presenting the data in a simple and clear form are essential for stakeholders to be able to create a beautiful environment together.

Third, the historical expression of the cultural landscape, such as the ancient camphor tree in the village, carries sentimental value for visitors. This emotion is deeper and richer than the general sense of 'beauty'. Therefore, special attention should be paid to the emotional impact of such historical landscape services on people. Designers should 'moderate design' and avoid 'excessive design' to avoid destroying the carrier of people's feelings.

## 7. Conclusions

This paper presents a visualisation-based technique applied to enhance visitor experience. This evaluated technique can be used to for landscape planning and design. We developed a visualisation framework and analysed the relationship among emotions (positive and negative emotions in user experience), landscape aesthetics (selected metrics: coherence, complexity, enclosure, legibility, hierarchy, which can be mapped on the entropy image), and landscape services (active and passive enjoyment) in detail. From the visualisation framework, we can conclude that the indicators related to landscape aesthetics and services generate different emotions throughout the tour. Users experience a feeling of novelty and thus positive emotions for rare landscapes, whereas when the landscape remains the same long periods of time, boredom and negative emotions are generated. The landscapes of the middle ground and background in the nodes' images must be focused while planning the type of landscape services, where a good atmosphere can generate positive emotions and a bad atmosphere can lead to negative emotions. Designers and planners tend to pay attention to the design of main objects in a specific project, and they typically ignore the landscapes around the main object which influence the tourists' experience.

Although the traditional visualisation method only shows a single aspect of information based on local landscape and geographical elements, this visualisation framework connects landscape aesthetics, landscape services, and user experience, thereby causing emotional component visualisation and showing multiple aspects of information. Emotion visualisation is the assessment tool and presentation method that can be utilised in rural design planning. The positive and negative emotions generated by users during tourism can inform and advise designers at different design stages to enhance user experience. The visualisation framework presented in this paper has general relevance, is not limited to a specific location, and can be used in landscape design, landscape planning, route planning, and landscape services.

This study selected an ancient village in western Zhejiang, China, as the design and planning sample to preserve the cultural landscape of the village and to showcase the local culture and characteristics to a large extent to visitors. The visualisation approach can be

effectively used to evaluate the design and planning project and obtain new strategies for the next design and planning around atmospheric landscapes.

In the future studies, researchers can continue to refine specific indicators, such as landscape aesthetics, landscape services, and user experience, and optimise them according to the specific experimental site. Simultaneously, techniques such as machine learning can added to refine local design planning paradigms, and a database can be developed to obtain planning and design for other areas. The visualisation method can accurately grasp the relationship between user experience and landscape planning, and design and provide better suggestions to designers and planners.

**Author Contributions:** Conceptualization, W.W., M.W. and K.O.; methodology, W.W., M.W. and K.O.; software, W.W.; validation, W.W., M.W., K.O. and D.Z.; formal analysis, W.W.; investigation, W.W.; resources, W.W. and D.Z.; data curation, W.W.; writing—original draft preparation, W.W.; writing—review and editing, W.W., M.W. and K.O.; visualization, W.W.; supervision, M.W. and K.O.; project administration, M.W. and D.Z.; funding acquisition, W.W. All authors have read and agreed to the published version of the manuscript.

**Funding:** This research received no external funding.

**Data Availability Statement:** The data presented in this study are available in article.

**Conflicts of Interest:** The authors declare no conflict of interest.

## Appendix A. Landscape Classification in Rural Places

**Table A1.** Landscape Classification in Rural Places and Examples.

| Landscape Classification | Examples |
|---|---|
| Productive landscape space (outdoor) | aquaculture system space, animal feeding space, botanic garden, plantation garden, large area landscape plant, agricultural productive landscape space, agricultural facility landscape space |
| Leisure space (outdoor) | square, pavilion, corridor, lawn, garden, greenhouse, market, gallery, stage |
| Artistic landscape | landscape sculpture, signboard, sign, lamplight |
| Public furniture | seat, table, garbage can, street lamp, peristele, fence, fitness equipment, sunny or rainy facilities, cook kit |
| Buildings | residential buildings, public buildings, productive buildings |
| Residential buildings | residence with courtyard |
| Public buildings | hotel, reception center, restroom, stadium, various exhibition halls, rental agency, parking lot, station, supermarket, laundry tray |
| Productive buildings | seed storage, livestock farm, food and feed processing station, tractor station |
| Road landscape | entrance to a village, street, lane, bridge, plank road, walkway, greenway |
| Water landscape | water gap, well bay, pool, ravine stream, sea |
| Animal landscape | mammal, amphibious animals, pets |
| Plants landscape | wood, bamboo forest, ancient trees, nature plants, landscape plants, courtyard plants |
| Matrix landscape | farmland, rock, sand beach, mountain, wood, lake, sea, land, terrace |

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
