# Peer review of "Exploring Visualisation Methodology of Landscape Design on Rural Tourism in China"

_buildings, doi:10.3390/buildings12010064_

Round 1

Reviewer 1 Report

Dear Authors,

Greetings.

Indeed, it gave me immense pleasure in reading your well written article and perfectly tuned content, I wish to provide some comments for you to look upon, please go through the below mentioned comments and fine tune your manuscript.

  1. In the title "Exploring a visualisation methodology" is that "a" required?
  2. I suggest authors to provide study area map without fail, please note that international readers will read this paper and they will be keen on understanding where this location is. Also provide Geolocation ( lat, long coordinates). Fig 1 seems to be representative image, please provide RSGIS based image.
  3. 3.4 Method design: Change it to: Design process or Design methodology
  4. Line 30, are they local tourists or foreign? 
  5. Line 33-35 rewrite please, it gives improper meaning
  6. Line 37: unsatisfactory to whom? Clearly write this sentence
  7. Line 53 are indispensable: is indispensable
  8. Line 55 : it can help determine: it can help to determine
  9. Line 56: cooperates is a single word
  10. Line 56-60, is putting chairs outside a facelift? Does it have ecological impact? or aesthetical impact?
  11. Line 72: but also through
  12. Line 83:noumenon: give alternate word please.
  13. Line 87 :t human activities and nonbiological and biological, change to human activities, biological and non biological.....
  14. Section 1.2 is too elaborate, optimise please, if this is a review paper, this section holds good but for this paper, please be on point.
  15. Line 137, why discussion on digital technology here, and that too what technology author meant for?
  16. Line 164-166 give citation
  17. Line 170: analyse a .....
  18. Line 175 et al computed....
  19. Section 1 too elaborate, please see to be on point and reduce unwanted lines, surely readability will increase. nearly 190 lines are there in this section.

Section 2:

  1. Line 197:is considered an. . is considered as an...
  2. Line 199 what is the difference between historic landscape and heritage? Is it heritage structures?
  3. Line 210:We selected the fourth enjoyment in the landscape service (LS) divided": We selected the fourth enjoyment in in the.. and divided....
  4. Line 218: Before designing and planning.
  5. Change the word " enjoyment" everywhere and replace it with suitable word like satisfaction etc.
  6. Line 221: historical or rural.

Section 3 and 4:

  1. Line 229, change summer resort to summer tourist spot
  2. Line 239: I am not seeing it in figure 3, is it correct?
  3. Line 258: instead of depending on photos? Which photos?
  4. Line 272 which scale authors used? Like Likert scale?
  5. Section 3.4, is it authors design or based upon literature?
  6. Fig 6 is it landscape aesthetics wrongly spelled? Its given as atheistic i don't know why. Same in line 345. but in line 357 its given as aesthetics, I got confused, please check
  7. Is figure 8 adopted or authors work?
  8. Line 422, we renovated, who? Use alternate words please
  9. Line 463, when the color of each node varies....
  10. Line 471 put a comma after sixth node
  11. Line 474 they are sold in market
  12. Line 476, fishes were fed...
  13. Line 494: Visitors already have complacency.. this will replace that sentence.
  14. Line 578-9 values of regression coefficient is not confirmalReferences
  1. I personally feel there are too many references in this paper, optimise it if its not of real use.
  2. Ref 1: The State Council The People’s Republic of China: give a comma as The State Council, The People’s Republic of China.
  3. Ref 5,9,14,15,23,25,41,42 need full details, check author guidelines for book and provide all details.
  4. Ref 7 and 6, are they both in same format? Follow same format for all references please
  5. Why authors used many books as references is not clear to me
  6. Ref 29,38 is it available for others to read?
  7. Is ref 34 really related to this study?
  8. I am unable to view ref 71 online
  9. Ref 74 why the word et al needed?

Author Response

Response to the Reviewer

Title: Exploring visualisation methodology of landscape design on rural tourism in China

Buildings- - 1523027

Dear Reviewer,

The co-authors and I would like to thank you for the time and effort spent in reviewing the manuscript. We would like to thank the reviewer for all the suggestions. We have carefully followed the reviewers' requests to answer each of the questions and comments. Each comment will be directly addressed regarding the modified manuscript with changes highlighted in yellow. The modified manuscript is in the attachment, please see the attachment.

Our response to the Reviewer‘s Comments as follows:

1.In the title "Exploring a visualisation methodology" is that "a" required?

Response 1: I changed the title : Exploring visualisation methodology of landscape design on rural tourism in China.

2.I suggest authors to provide study area map without fail, please note that international readers will read this paper and they will be keen on understanding where this location is. Also provide Geolocation ( lat, long coordinates). Fig 1 seems to be representative image, please provide RSGIS based image.

Response 2: I add the links to google map, and noted the Geolocation on the map to show more details.

3.3.4 Method design: Change it to: Design process or Design methodology

Response 3: I changed the subheading to the 4 Method.Line 229,4.1 Design process, 4.2Visualisation framework.

4. Line 30, are they local tourists or foreign?

Response 4: They are local tourists who travel to rural places. I add the explanation: line31-32“the total number of local tourists who travel in the rural places…”

5. Line 33-35 rewrite please, it gives improper meaning

Response 5: I deleted these sentences because they were ambiguous.

6. Line 37: unsatisfactory to whom? Clearly write this sentence

Response 6: I deleted these sentences because they were too long.

7. Line 53 are indispensable: is indispensable

Response7: Line 78, I revised this word.

8. Line 55 : it can help determine: it can help to determine

Response8: Line 80, I revised this sentence.

9. Line 56: cooperates is a single word

Response9: Line 81, I revised this word.

10. Line 56-60, is putting chairs outside a facelift? Does it have ecological impact? or aesthetical impact?

Response10: I deleted this case because It has little to do with the article.

11. Line 72: but also through

Response11: Line 96, I revised this word.

12.Line 83:noumenon: give alternate word please.

Response 12: I deleted these sentences because this paragraph was too long. I deleted Table1. Ecosystem service classification, because it was some repetition with the text that follows.

13. Line 87 :t human activities and nonbiological and biological, change to human activities, biological and non biological.....

Response13: Line 101,I revised this sentence.

14. Section 1.2 is too elaborate, optimise please, if this is a review paper, this section holds good but for this paper, please be on point.

Response14: I revised the points in the introduction and revised the previous 1.1-1.3 to chapter2 as Literature review and reduced some contents, which could be more on point.

15. Line 137, why discussion on digital technology here, and that too what technology author meant for?

Response15:I deleted this sentence here.

16. Line 164-166 give citation

Response16:I deleted the this part, because it was some repetition with the text that follows.

17. Line 170: analyse a .....

Response17: Line 158, I revised this word.

18. Line 175 et al computed....

Response18: Line 163, I revised this word.

19. Section 1 too elaborate, please see to be on point and reduce unwanted lines, surely readability will increase. nearly 190 lines are there in this section.

Response19: I revised the points in the introduction and revised the previous 1.1-1.3 to the chapter2 Literature review.

Section 2:

1. Line 197:is considered an. . is considered as an...

Response 1: Line 127,I revised this word.

2. Line 199 what is the difference between historic landscape and heritage? Is it heritage structures?

Response 2: Line 130. Yes, it is, the heritage has more cultural values for the specific place or human beings, they are always be protected by local government.

3. Line 210:We selected the fourth enjoyment in the landscape service (LS) divided": We selected the fourth enjoyment in the.. and divided....

Response 3: 268-269,I revised this sentence.

4. Line 218: Before designing and planning.

Response 4: I deleted this paragraph because it was some repetition with the other parts.

5. Change the word " enjoyment" everywhere and replace it with suitable word like satisfaction etc.

Response 5: Enjoyment is the part of classification in landscape services,it was divided as passive enjoyment and active enjoyment. I gave more explanation in the 4.2visualisation framework.

6. Line 221: historical or rural.

Response 6: I deleted this paragraph because it was some repetition with the other parts.

Section 3 and 4:

1. Line 229, change summer resort to summer tourist spot

Response 1: Line 179, I revised this sentence.

2. Line 239: I am not seeing it in figure 3, is it correct?

Response 2: Yes, it is. I revised this mistake. On line194, it is figure2.

3. Line 258: instead of depending on photos? Which photos?

Response 3: Line 203-205, I revised this sentence.

4. Line 272 which scale authors used? Like Likert scale?

Response 4: Line 218-219. Yes, it is the Likert scale.

5. Section 3.4, is it authors design or based upon literature?

Response 5:It was mostly the authors' design but also based upon the literature.

6. Fig 6 is it landscape aesthetics wrongly spelled? Its given as atheistic i don't know why. Same in line 345. but in line 357 its given as aesthetics, I got confused, please check.

Response 6:Line302, I revised “aesthetics” in the figure. Line263, and add the visualization framework as Fig 5 to explain more about the method.

7. Is figure 8 adopted or authors work?

Response 7:Figure9,line364,It was the author's work based on the entropy images.

8. Line 422, we renovated, who? Use alternate words please

Response 8: I deleted this sentence because it wasn’t clear enough.

9. Line 463, when the color of each node varies....

Response 9:Line 408, I revised this sentence.

10.Line 471 put a comma after sixth node

Response 10: Line 416, I put a comma after the sixth node.

11. Line 474 they are sold in market

Response 11: Line 419, I revised this word.

12. Line 476, fishes were fed...

Response 12: Line 421, I revised this word.

13. Line 494: Visitors already have complacency.. this will replace that sentence.

Response 13: Line 439-440, I revised this sentence.

14. Line 578-9 values of regression coefficient is not confirmal

Response 14: Line 523, I revised the “blue areas” word according to Table7.

References

As for the references, we deleted some references and revised many of them, please check them.

1. I personally feel there are too many references in this paper, optimise it if its not of real use.

Response 1: I revised this Reference, and declined the numbers from 74 to 58.

2. Ref 1: The State Council The People’s Republic of China: give a comma as The State Council, The People’s Republic of China.

Response 2: I revised this sentence.

3.Ref 5,9,14,15,23,25,41,42 need full details, check author guidelines for book and provide all details.

Response 3: I checked the author guidelines and provide all details.

4. Ref 7 and 6, are they both in same format? Follow same format for all references please

Response 4:I’m sorry, some references don’t have the DOI, what should I do?

5. Why authors used many books as references is not clear to me

Response 5: I revised the books,and deleted some of them.

6. Ref 29,38 is it available for others to read?

Response 6: I deleted 29 and 38.

7. Is ref 34 really related to this study?

Response 7: I deleted 34 and related parts.

8. I am unable to view ref 71 online

Response 8: I deleted 71 and related parts.

9. Ref 74 why the word et al needed?

Response 9: I revised 74.

Reviewer 2 Report

Dear authors,

Thank you for sharing your research with us. In my opinion, the topic of the paper is interesting and there are some minor points that should be addressed:

  • Introduction is missing - it should contain the main purpose and the scientific contribution needs to be made clear (as well as in the conclusion). It is necessary to identify and highlight the gap in the literature that the authors are aiming to bridge. Subchapters 1.1. up to 1.3. should make a new chapter, not the Introduction.
  • It would be advisable to include more information on the survey itself (i.e. time period when the survey was conducted, how it was conducted, how the sample was defined, etc.).
  • Research limitations should be inserted and explained.

Good luck!

Author Response

Response to the Reviewer

Title: Exploring visualisation methodology of landscape design on rural tourism in China

Buildings- - 1523027

Dear Reviewer,

The co-authors and I would like to thank you for the time and effort spent in reviewing the manuscript. We would like to thank the reviewer for all the suggestions. We have carefully followed the reviewers' requests to answer each of the questions and comments. Each comment will be directly addressed regarding the modified manuscript with changes highlighted in yellow. The modified manuscript is in the attachment, please see the attachment.

Our response to the Reviewers Comments as follows:

1.Introduction is missing - it should contain the main purpose and the scientific contribution needs to be made clear (as well as in the conclusion). It is necessary to identify and highlight the gap in the literature that the authors are aiming to bridge. Subchapters 1.1. up to 1.3. should make a new chapter, not the Introduction.

Response 1:I revised the points in the introduction and revised the previous 1.1-1.3 to chapter2 as a Literature review. I rewrote the introduction and conclusion to match the points of the research, please check them.

2.It would be advisable to include more information on the survey itself (i.e. time period when the survey was conducted, how it was conducted, how the sample was defined, etc.).

Response 2:I add more information in 3.3, 4.1 Design Process. I revised the visualization framework to 4.2. And add Figure 5. Flowable of the visualization method framework to explain more about the method.

3.Research limitations should be inserted and explained.

Response 3:I revise the research limitations in 6. Discussion, please check them.

Reviewer 3 Report

The paper examines the problem of perception of rural tourism with due thoroughness and makes meaningful suggestions for both theoretical and practical landscape planning. The research is based on in-depth and detailed processing and the visual illustrations adequately reflect the logic of the research.

Compared to the main analysis of the paper, the wording of the conclusion is rough. It is proposed to revise the conclusion, in particular not to change the content of the conclusions, but to raise the wording and formulation to a higher level of quality.

Author Response

Response to the Reviewer

Title: Exploring visualisation methodology of landscape design on rural tourism in China

Buildings- - 1523027

Dear Reviewer,

The co-authors and I would like to thank you for the time and effort spent in reviewing the manuscript. We would like to thank the reviewer for all the suggestions. We have carefully followed the reviewers' requests to answer each of the questions and comments. Each comment will be directly addressed regarding the modified manuscript with changes highlighted in yellow. The modified manuscript is in the attachment, please see the attachment.

Our response to the Reviewers Comments as follows:

Point: Compared to the main analysis of the paper, the wording of the conclusion is rough. It is proposed to revise the conclusion, in particular not to change the content of the conclusions, but to raise the wording and formulation to a higher level of quality.

Response:

I rewrote the introduction and conclusion to match the points of the research; as for the conclusion, I revised some points, please check them.
